# Anticancer Effects of Fucoxanthin through Cell Cycle Arrest, Apoptosis Induction, Angiogenesis Inhibition, and Autophagy Modulation

**DOI:** 10.3390/ijms232416091

**Published:** 2022-12-17

**Authors:** Shade’ A. Ahmed, Patricia Mendonca, Rashid Elhag, Karam F. A. Soliman

**Affiliations:** 1Division of Pharmaceutical Sciences, College of Pharmacy and Pharmaceutical Sciences, Institute of Public Health, Florida A&M University, Tallahassee, FL 32307, USA; 2Department of Biology, College of Science and Technology, Florida A&M University, Tallahassee, FL 32307, USA

**Keywords:** fucoxanthin, carotenoids, cancer, apoptosis, autophagy

## Abstract

Cancer accounts for one in seven deaths worldwide and is the second leading cause of death in the United States, after heart disease. One of the standard cancer treatments is chemotherapy which sometimes can lead to chemoresistance and treatment failure. Therefore, there is a great need for novel therapeutic approaches to treat these patients. Novel natural products have exhibited anticancer effects that may be beneficial in treating many kinds of cancer, having fewer side effects, low toxicity, and affordability. Numerous marine natural compounds have been found to inhibit molecular events and signaling pathways associated with various stages of cancer development. Fucoxanthin is a well-known marine carotenoid of the xanthophyll family with bioactive compounds. It is profusely found in brown seaweeds, providing more than 10% of the total creation of natural carotenoids. Fucoxanthin is found in edible brown seaweed macroalgae such as *Undaria pinnatifida, Laminaria japonica,* and *Eisenia bicyclis*. Many of fucoxanthin’s pharmacological properties include antioxidant, anti-tumor, anti-inflammatory, antiobesity, anticancer, and antihypertensive effects. Fucoxanthin inhibits many cancer cell lines’ proliferation, angiogenesis, migration, invasion, and metastasis. In addition, it modulates miRNA and induces cell cycle growth arrest, apoptosis, and autophagy. Moreover, the literature shows fucoxanthin’s ability to inhibit cytokines and growth factors such as TNF-α and VEGF, which stimulates the activation of downstream signaling pathways such as PI3K/Akt autophagy, and pathways of apoptosis. This review highlights the different critical mechanisms by which fucoxanthin inhibits diverse cancer types, such as breast, prostate, gastric, lung, and bladder development and progression. Moreover, this article reviews the existing literature and provides critical supportive evidence for fucoxanthin’s possible therapeutic use in cancer.

## 1. Introduction

Worldwide, one in seven deaths is due to cancer, and in the United States, cancer is the second leading cause of death besides cardiovascular disease. In 2022, 1,918,030 cancer cases and 609,360 cancer deaths are anticipated to occur in the United States [1]. The World Health Organization (WHO) estimates that cancer is the leading cause of death, with a global 10 million deaths in 2020. The most commonly diagnosed cancers in the United States are lung, breast, and prostate cancers, accounting for more than 50% of all new cancer cases [2]. In 2022, it is estimated that 287,850 women will be diagnosed with breast cancer, making it the most diagnosed cancer. Prostate cancer is the leading cancer diagnosed in men, with 268,490 expected cases this year. Lung cancer is the third most diagnosed cancer, with 236,740 new cases this year [3]. 

Many therapeutic options for cancer treatment are available, typically including radiotherapy, chemotherapy, and surgery. Cancer cells show resistance to chemotherapy in many cases of cancer treatment. Currently, more focus is directed on targeted and individualized therapy. Extensive research focuses on potential approaches to treat cancer using diverse signaling pathways, multidrug resistance, cell cycle checkpoints, and anti-angiogenesis. Moreover, modulation of the tumor microenvironment (TME) has been targeted for cancer therapy since the TME is crucial for tumor development and progression. The TME is diverse and extremely complex, allowing cancer cell proliferation and reprogramming of cancer cells to acclimatize to changes within TME [4]. Moreover, tumor cell growth, apoptosis suppression, immune system inhibition, and immune surveillance avoidance are linked to TME [5]. Furthermore, the activation or deactivation of tumor suppressor genes or oncogenes can activate apoptosis and uncontrolled cell cycle progression [6]. 

Therefore, novel therapeutic approaches are necessary to treat cancer patients. Natural products have exhibited anticancer effects that may be beneficial in treating cancer, having fewer side effects, low toxicity, and affordability. Studies of both in vitro and in vivo models have revealed that many natural compounds possess the capability to suppress angiogenesis, inhibit proliferation and induce autophagy and apoptosis [7]. Several of these natural compounds can treat cancer and lower cancer incidence at an affordable and reasonable price [8]. Numerous research investigations indicated that various edible plants contain carotenoids, the bioactive compounds that prevent cancer [9]. Carotenoids’ natural orange-red food pigment is found in many fruits and vegetables, such as sweet potatoes, squash, melons, cantaloupe, tangerines, papaya, pumpkin, tomatoes, and carrots [10]. Carotenoids are anticarcinogenic, which contribute to the inhibition of malignant transformation, tumorigenesis, and antioxidation [11]. Natural compounds used as anticancer agents are usually less toxic, affordable, and present anti-inflammatory, antiproliferative, antiangiogenic, and antioxidant properties. Moreover, extracts isolated from marine organisms have been found to have potent anticancer activities [12]. Fucoxanthin is one of these carotenoids with anticancer potential. Fucoxanthin has several anticancer effects, including anti-proliferation, cell cycle arrest, pro-apoptosis, anti-metastasis, and tumor inhibition [13]. Thus, the current review focuses on fucoxanthin’s molecular mechanisms to inhibit tumor development and progression, providing evidence of its potential option to treat/prevent cancer.

## 2. Cancer Statistics

In the United States, breast cancer is the second most frequent cancer, following lung cancer [14]. The 5-year survival rate for women with breast cancer is 90%, and the 10-year survival rate for women with non-metastatic invasive breast cancer is 84% [15]. Lung cancer is one of the leading causes of cancer, making up 25% of all cancer deaths [16]. According to the Global Burden of Disease study, the average 5-year survival rate for lung cancer is 17.7% [17]. GLOBCAN 2020 statistics indicate that gastric cancer caused 800,000 deaths, accounting for 7.7% of all cancer deaths, and is one of the global health challenges [18]. Moreover, the American Cancer Society data in 2022 shows that about 81,180 new cases of bladder cancer and 17,100 deaths from bladder cancer are expected to occur. Bladder cancer is the fourth most common cancer in men and is less common in women. The 5-year survival rate of people with bladder cancer is 77% [19]. In 2022, 34,500 deaths and 268,490 new cases of prostate cancer are expected in the United States [20]. The 5-year survival rate for men with prostate cancer is 98% [21]. On the other hand, melanoma is the third most common skin cancer and the leading cause of skin cancer death in the united states [22]. The 5-year survival rate for patients with cutaneous melanoma is 26.4% [23]. In 2022, The American Cancer Society approximates that there will be around 60,650 new cases and 24,000 deaths due to leukemia in the United States [24]. The 10-year survival rate for patients with leukemia is 87% [25]. There are 1.93 million new cases of colorectal cancer, and 940,000 cases caused death in 2020 worldwide. New colorectal cancer cases are expected to reach 3.2 million by 2040 [26]. The 5-year survival rate of colorectal cancer is 64.4% [27]. The World Health Organization (WHO) estimates that cancer is the leading cause of death, with a global 10 million deaths in 2020 [28]. Table 1 displays cancer types and survival rates with their respective percentages.

## 3. Treatment Options for Cancer

Treatment options for cancer include targeted therapy, immunotherapy, hormone therapy, radiation therapy, surgery, and chemotherapy [29]. The leading trending approach in treating localized cancer is costly surgery. In addition, radiation therapy can reduce the risk of local cancer recurrence and eliminates cancer cells not seen during surgery; however, there are many side effects [30]. Side effects of radiation therapy include peeling, redness, itching, and soreness, leaving skin moist and prone to infection [31]. Radiation therapy can treat gastric cancer and has a 5-year survival rate of 23% [32]. A study showed that after a follow-up of 35 years after skin hemangioma radiation therapy, the risk of developing other melanoma was 2.5 folds higher [33]. 

Chemotherapy, one of cancer’s standard therapies, can be administered intramuscularly or intravenously, or orally to reach the bloodstream [34]. The administration of chemotherapeutic drugs with various molecular targets has provided advantages such as improvement of adverse effects and efficacy progress. However, the search for natural-based options with less toxicity has become imperative [35]. Targeted therapy causes less harm to normal cells than radiation therapy and chemotherapy, but it may cause targeted therapy resistance. Targeted therapy resistance has been shown in various cancers such as lung, leukemia, colorectal, and gastric [36,37,38,39]. Bevacizumab and Ramucirumab are monoclonal antibodies used as targeted therapy to treat lung cancer. KRAS inhibitors are used as immunotherapy to prevent KRAS gene mutations and cellular growth of lung cancer. Side effects of using monoclonal antibodies and KRAS inhibitors for lung cancer treatment include serious side effects such as severe bleeding, blood clots, liver damage, slow wound healing, and perforations in the intestine [40]. Therapeutic antibody pembrolizumab is used as immunotherapy to treat various cancers such as lymphoma, lung, melanoma, and gastric cancer and has a 5-year survival rate of 23.2% [41]. Side effects of pembrolizumab include hepatitis, thyroid dysfunction, and pneumonitis, which can lead to the end of treatment [42]. 

In breast cancer, hormone therapy is a treatment that stops the growth of cancer which uses hormones [43]. Hormone therapy with tamoxifen or estrogens can affect the body and may increase or cause endometrial cancer [44]. Luminal A and luminal B breast tumors are often treated with tamoxifen, which has been shown to promote stemness and contribute to metastasis and hormone therapy resistance [45]. Patients with luminal A and luminal B type cancers who take tamoxifen have a 5-year survival rate of 76% [46]. HER2-positive breast tumors test positive for the human epidermal growth factor receptor 2 [47]. Trastuzumab is a humanized murine IgG monoclonal antibody that binds the HER-2 receptor and is often used for treatment. Although trastuzumab impacts HER-2-positive breast cancer, trastuzumab causes resistance in HER-2-positive tumors [48]. Patients with HER-2 tumors who take trastuzumab have response rates of 11–26% and have shown to not be responsive to this therapy [49]. Due to many issues with current treatments, such as radiation therapy, immunotherapy, surgery, chemotherapy, targeted therapy, and hormone therapy, there is a need for effective treatments for cancer. Therefore, natural compounds would be an alternative, less toxic, and less expensive cancer treatment [50].

## 4. Natural Products

Diet and nutrition are known to be effective and preventative strategies for cancer. Numerous natural dietary products have been shown to play a potential role in cancer prevention and treatment [51]. Phytochemicals are bioactive nutrient chemicals found in vegetables, fruits, grains, or other plant foods that offer desirable health benefits and have great potential in reducing chronic diseases such as heart disease, obesity, diabetes, and lung, breast, and prostate cancer [51,52,53,54]. Phytochemicals include flavonoids, carotenoids, terpenes, phytoestrogens, stanols, or phenolic acids. Flavonoids are contained in vegetables, fruits, fungi, chocolates, wine, and teas and can potentially prevent cancer cell growth [55]. Fat-soluble carotenoids are found in orange, yellow, and red fruits, seaweed, or dark leafy vegetables and possess strong cancer-fighting properties [56]. Terpenes in citrus fruits can slow cancer cell growth and prevent virus-related illnesses [57]. Phytoestrogens are found in grapes, berries, plums, soybeans, tofu, and garlic and may lower the risk for osteoporosis, heart disease, menopausal symptoms, and breast cancer [58]. Stanols found in nuts, grains, and legumes may reduce the possibility of cardiovascular diseases and stroke and lower blood cholesterol levels [59]. Phenolic acids are found in coffee, fruits, vegetables, nuts, cereals, legumes, oilseeds, and herbs. They can promote anti-inflammatory processes in the body and may prevent cellular damage due to free-radical oxidation reactions [60].

Numerous studies proposed that consuming fruits (particularly cruciferous vegetables) and soy products reduces breast, lung, ovarian, pancreas, and prostate cancer risk [61,62,63]. Multiple investigations indicate that specific natural products are identified as cancer chemoprevention and are used to slow, suppress, prevent, or reverse carcinogenic activities. These compounds in breast, prostate, gastric, and liver cancer display anti-inflammatory, antiproliferative, anti-metastatic, ant-angiogenic, and apoptotic properties [64,65,66,67]. High intake of some natural dietary products may reduce the risk of recurrence and improve survival in breast cancer patients [68]. Extensive experimental studies indicated that many nutritional natural products, including pomegranate, edible macrofungi, marine macro- and micro-algae, curcumin, teas, cereals, citrus fruit, grape, mango, and spice, shows to affect the development and progression of cancer (Figure 1).

Fruits are a rich source of great antioxidants and contain high polyphenols contents, which help reduce cancer risk [69,70,71,72,73]. Meanwhile, cruciferous vegetables such as cauliflower, broccoli, and brussels sprouts are a rich source of isothiocyanates. Isothiocyanates have been attributed to have chemopreventive activities for cancer [74]. Moreover, spices such as ginger and garlic have been used for dietary and medicinal purposes since ancient times. Ginger extract inhibits proliferation and induces the G2/M phase and apoptosis in colon cancer [75]. Garlic is known to inhibit cell growth and induced apoptosis and cell cycle arrest in G0/G1 phase in gastric cancer [76]. In addition, edible macrofungi offer anti-breast, liver, lung, and colorectal cancer effects, such as the induction of apoptosis, inhibition of proliferation, and suppression of angiogenesis [77,78,79]. Moreover, grape seed extract could induce cell cycle arrest at the S phase [80]. In addition, marine macro- and micro-algae seaweeds possess anti-metastatic and induce apoptosis of cancers in general [81].

### 4.1. Carotenoids and Fucoxanthin

Carotenoids are a subfamily of tetraterpenoids, or isoprenoids, synthesized by photosynthetic organisms, including microalgae, macroalgae, plants, and fungi. Carotenoids’ natural orange-red food pigment is found in many fruits and vegetables, such as sweet potatoes, squash, melons, cantaloupe, tangerines, papaya, pumpkin, tomatoes, and carrots [82]. β-carotene, β-cryptoxanthin, zeaxanthin, lutein, lycopene, and α-carotene are the predominant carotenoids in the diet, including 90% of circulating carotenoids and are found to be anticarcinogenic, which contribute to inhibition of malignant transformation, tumorigenesis, and antioxidation [83]. Carotenoids are also grouped into provitamin A, non-provitamin A. α -carotene, β-cryptoxanthin, and β-carotene are provitamin A carotenoids. Zeaxanthin, lycopene, and lutein are non-provitamin A [84]. Carotenoids are divided into two classes: carotenes and xanthophylls. Carotenes are classified as hydrocarbons which are isoprene derivatives, and xanthophylls are oxygen derivatives of carotenes. In color, carotenes are more orange, while xanthophylls are more yellow due to different absorption of light [85]. One of the most well-studied xanthophylls is fucoxanthin [86].

Fucoxanthin is a carotenoid of the xanthophyll family, produced by marine organisms such as macroalgae of the genus Fucus and microalgae such as *Phaeodactylum tricornutum*. Fucoxanthin (Figure 2) is an orange xanthophyll pigment derived from brown algae and microalgae. Fucoxanthin is one of the most abundant carotenoid pigments, contributing more than 10% of the estimated total carotenoid production in nature, especially in the marine environment. One of the most common food ingredients in East Asian countries such as Japan is seaweed. Seaweed is a source of fascinating natural compounds that have been utilized for medicinal purposes for thousands of years. Attention toward natural products isolated from seaweed is due to their source of bioactive compounds and their health benefits [87]. 3′-acetoxy-5,6-epoxy-3,5′-dihydroxy-6′,7′-didehyro-5,6,7,8,5′,6′-hexahydro-β, β-carotene-8-one (fucoxanthin) is a pigmented, orange-colored xanthophyll was first isolated by Page and Willstätter in 1914 [88]. (Figure 2).

The molecular weight of fucoxanthin is 658.9 g/mol (C_42_H_58_O_6_), and it absorbs green and blue light at 450–540 nm, conveying an olive-brown color to algae. Fucoxanthin’s pKa value is 14.47, which indicates that it is weakly acidic. [90]. Fucoxanthin is vulnerable to light, oxygen, thermal deprivation, and pH because of its numerous conjugated double bonds, which can be an issue for prolonged storage periods. Studies have established the need for fucoxanthin encapsulation [91]. Fucoxanthin has an exclusive molecular structure that displays an allenic bond, a 5,6-monoepoxide, a heavily conjugated system, and some epoxy, hydroxy, carbonyl, epoxy, hydroxyl, carboxyl functional groups. Due to its structure and chirality, it is unstable and affected by air light and heating. However, due to these features, fucoxanthin has high antioxidant activity [92]. 

Fucoxanthin has demonstrated phytonutrient benefits, including anti-inflammatory, anti-diabetic, antiobesity, and anticancer effects. Fucoxanthin shows antiobesity effects in animal models of type 2 diabetes through the uncoupling protein (UCP) 1 expression in white adipose tissue (WAT) of KKAy mice [93]. Studies show that fucoxanthin has been reported to penetrate the blood–brain barrier. It is exceptional from other carotenoids to exercise anti-neurodegenerative disorder effects against oxidative stress, amyloid protein aggregation, neurotransmission dysregulation, and gut microbial disorder [94].

### 4.2. Bioavailability and Metabolism of Fucoxanthin

Fucoxanthin is hydrolyzed into fucoxanthinol by digestive enzymes such as lipase in the gastrointestinal tract and taken up by intestinal cells. Fucoxanthinol is viewed as the essential dynamic metabolite in humans. Absorption and metabolism of fucoxanthin are related to its bioavailability, the negligible part of the drug that arrives in systemic circulation [95]. The two principal metabolites of fucoxanthin are fucoxanthinol and amarouciaxanthin A (Figure 3). The absorption rate of fucoxanthin is affected by lipids. Upon ingestion, fucoxanthin is broken down to fucoxanthinol in the gastrointestinal tract through hydrolysis by digestive enzymes such as cholesterol esterase and lipase. Fucoxanthinol is then used further and converts to amarouciaxanthin A in the liver [96]. Experimental studies have shown that the utilization of fucoxanthin promotes metabolism. However, this metabolic boost did not stimulate the central nervous system [97]. After a day-to-day intake of stir-fried wakame (6 g dry weight) and 6.8 mg (9.26 μmol) of fucoxanthin for 1 week, 0.8 pmol/mL of fucoxanthinol was detected in human plasma [98]. The utilization of fucoxanthin as a nutraceutical in nutrient supplements and functional food is low due to its poor stability, poor water solubility, and limited bio-accessibility [99].

### 4.3. Fucoxanthin Toxicity 

Fucoxanthin has been confirmed to be a safe carotenoid with no side effects at 0.5% w/v in human skin and at 20–2000 mg /kg body weight in rodents [101]. Toxicity studies conducted on rats for 4 weeks, with oral dosing of fucoxanthin, showed no harmful adverse effects after daily treatments. Moreover, in vivo investigations with fucoxanthinol showed no significant adverse effects [102].

## 5. Anticancer Mechanisms of Fucoxanthin

### 5.1. Cell Proliferation and Cell Cycle Arrest Mechanisms

Cell cycle arrest is the stopping point of the cell cycle where it is no longer involved in replication and division processes [103]. Cell proliferation is a rapid cell population expansion due to cell division and growth and involves distinctive stages [104]. Cells in the resting G0 phase are stimulated to enter the cell in the G1 phase, and during this time, the cell prepares for the S phase or DNA synthesis. Management of cellular proliferation occurs in the G1 phase of the cell cycle division cycle. Entering the S phase from the G1 phase symbolizes a committed mechanism to complete the cell cycle and divide in a tightly controlled process [105]. The second phase of inactivity G2 phase and the arrangement for separation of chromatids in the mitotic M phase follow the S phase [106].

Chromosomal instability, genomic instability, and unscheduled proliferation aid in the misregulation of cyclin-dependent kinases (CDKs) [107]. The DNA damage checkpoint protects cells from genotoxic agents such as ionizing radiation, free radicals, and chemicals, which can induce alterations in the DNA molecule. Cyclins are created and dismantled during the cell cycle and aid in regulating kinase activity. Many CDK types drive the cell cycle: CDK1, CDK2, CDK3, CDK4, CDK5, CDK6, CDK7, CDK8, CDK9, CDK10, CDK11, and CDK12 [108]. 

Checkpoints monitor the order and scheduling of the cell cycle, permitting proliferation. Systematic checkpoints and the cell cycle can be disrupted by malfunctioning genes or proteins that cause malignant cell alterations, leading to cancer [109]. Cell cycle arrest through changes in CDK activity is induced by the activation of checkpoints [110]. The inactivation of tumor suppressors and the activation of cyclins and cyclin-dependent kinases (CDK) are distinguishing qualities of cancer. This cellular process represents an assuring therapeutic target for cancer, primarily through cancer cell proliferation [111].

### 5.2. The Effect of Fucoxanthin on Cell Proliferation and Cell Cycle Arrest 

Fucoxanthin antiproliferative effects have been reported in several cancer cell lines, including T-cell leukemia, Hodgkin’s lymphoma, Burkitt’s lymphoma, melanoma cell lines, colon adenocarcinoma, and prostate cancer cells [112,113,114]. The antiproliferative effects of fucoxanthin have been confirmed in vivo by studies demonstrating tumor growth arrest and have been demonstrated in various cancer cell lines [115]. Fucoxanthin suppresses cell proliferation via cancer cell signaling pathways such as Akt/mTOR/S6 kinase signaling pathway in ovarian cancer cells. Fucoxanthin has displayed antiproliferative effects in prostate cancer (DU145, PC-3, LNCaP), urinary bladder cancer (EJ-1), melanoma (B16F10), breast cancer (MCF-7), and gastric cancer (MGC-803) [116]. 

Cell proliferation inhibition by fucoxanthin is due to cell growth arrest at the cell cycle’s G1 or G0/G1 phase. Fucoxanthin-induced G1 arrest in DU145 and HepG2 cells [117]. Fucoxanthin induced apoptosis and cell cycle arrest at the G2/M phase in gastric cancer MGC-803 cells [118]. Fucoxanthin increased cells in the G1 phase of the cell cycle. This is associated with increased expression of p27Kip1 and p15INK4B. Studies show that 10 μM of fucoxanthin leads to minor antiproliferative-inducing effects on MCF-7 and MDA-MB-231 cells [119]. In T24 cells, fucoxanthin inhibited proliferation in a time- and dose-dependent manner, initiating growth arrest at the G0/G1 phase of the cell cycle promoted by the depletion of cyclin D1, cyclin E, CDK-2, and CDK-4 [120]. Flow cytometry showed that treatment of human lymphatic endothelial cells (HLEC) with fucoxanthin (25, 50, 100 μmol/L) for 24 h resulted in cell cycle arrest in the S phase of MDA-MB-231 cells [121]. Fucoxanthin has also been shown to decrease cyclin D1 expression by altering GADD45a expression in HepG2 cells [122].

#### 5.2.1. The Intrinsic and Extrinsic Pathways of Apoptosis 

The process of apoptosis, or programmed cell death, is considered a key factor in processes involving proper development and function of the immune system, embryonic development, chemically induced cell death, and normal cell turnover [123]. Cancer development is controlled by a balance between cell proliferation and apoptosis. It is important to establish a tumor’s growth or regression in response to treatment. During carcinogenesis in epithelial tissue, genetic mutations assemble, and losses of cellular function can occur [124]. Understanding the pathogenesis of conditions resulting from disordered apoptosis can help develop or discover drugs that target specific apoptotic genes or pathways in cancer [125]. Caspase’s function is essential in the mechanism of apoptosis, and are both initiators and executioners. Two well-known pathways are the intrinsic (mitochondrial) and extrinsic (death receptor) pathways of apoptosis (Figure 4). The extrinsic death receptor pathway is activated as death ligands, such as tumor necrosis factor (TNF)-α, Fas ligand (FasL), and TNF-α -related apoptosis-inducing ligand (TRAIL), bind to a death receptor. The most common types of death receptors are type 1 TNF receptor (TNFR1) and FasL. Other death receptors, such as TNF receptor 2 (TNFR2), and death receptors 3-6 (DR3-6), can transmit death signals from the extracellular microenvironment to the cytoplasm [126]. 

TNFR1 and FasL have an intracellular death domain that recruits adapter proteins such as the TNF receptor-associated death domain (TRADD), the Fas-associated death domain (FADD), and caspase-8. When a death ligand binds to a death receptor, a binding site complex known as the death-inducing-signaling complex (DISC) is formed. Once DISC is activated, then it initiates the activation of pro-caspase-8. Caspase-8, an activated pro-caspase-8, is an initiator caspase and initiates apoptosis by cleaving other executioner or downstream caspases [127]. Caspase-8 cleaves Bid into tBid and initiates the mitochondrial apoptosis pathway or intrinsic mitochondrial pathway [128]. The activation of Bid compels cleavage by caspase-8, then the subsequent tBid relocates to the mitochondria and stimulates an increase in outer mitochondrial permeability preceding the release of apoptogenic proteins [129]. 

The intrinsic mitochondrial pathway of apoptosis is initiated intracellularly. This is due to increased mitochondrial permeability and the release of pro-apoptotic molecules such as cytochrome-c into the cytoplasm and other apoptotic factors such as second mitochondria-derived activator of caspase (Smac), apoptosis-inducing factor (AIF), Omi/high-temperature requirement protein A (HtrA2) and direct IAP Binding protein with Low Pi (DIABLO) [130]. Anti-apoptotic proteins such as Bcl-2 regulate apoptosis by blocking cytochrome c from being released from the mitochondria. The balance between anti-apoptotic and pro-apoptotic proteins determines the initiation of apoptosis. The binding of Omi/HtrA2 or Smac/DIABLO to the inhibitor of apoptosis proteins (IAPs) stimulates the activation of caspases. This facilitates disruption of the interaction of IAPs with caspase-3 or 9. The cytoplasmic release of cytochrome c activates caspase-3 by forming an apoptosome composed of Apaf-1, cytochrome c, and caspase-9 [131]. Intrinsic and extrinsic apoptosis pathways play an essential role in cancer treatment as it is a popular target of many treatment strategies [132].

#### 5.2.2. The Effect of Fucoxanthin on Apoptosis

A wide variety of natural products are recognized for their ability to induce apoptosis [133]. The effect of fucoxanthin has been investigated in many cancer cells. In colon cancer, Caco-2 cells, fucoxanthin has been shown to reduce Bcl-2 expression remarkably. Fucoxanthin treatment increased cleavages of procaspase-3 without any effect on the protein levels of antiapoptotic Bcl-2, Bcl-xL, or proapoptotic Bax in human leukemia cancer cells [134]. In lymphoma cancer cells, fucoxanthin was shown to increase caspase-dependent apoptosis and inhibit activation of AP-1, Akt, and NF-κB, while downregulating antiapoptotic proteins XIAP and Bcl-xL [113]. In non-small cell lung cancer, fucoxanthin modulated p53, upregulated modulator of apoptosis (PUMA), Fas, p21, and p53, and increased Bcl-2 and caspase-3 and -8 expression [135]. In the B16F10 human melanoma cell line, fucoxanthin downregulates Bcl-xL, activating caspase-3, -9, and PARP [136]. In prostate cancer PC-3 cells, fucoxanthin induced procaspase-3 and PARP cleavages and reduced Bcl-2 and Bax expression [137].

The induction of apoptosis is considered an appealing strategy for cancer therapy [138]. In breast cancer cell lines, fucoxanthin has induced apoptosis and protects against DNA damage. Fucoxanthin (25 µM) induces apoptosis in MCF-7 human breast cancer cells [139]. In vivo, fucoxanthin-induced apoptosis was accompanied by the downregulation of protein levels of Bcl-xL, resulting in the activation of caspase-3, caspase-9, and PARP in Balb/c mice [140]. Understanding the pathogenesis of conditions resulting from disordered apoptotic disorders will aid in the development or discovery of drugs that target specific apoptotic genes or signaling pathways. 

Figure 4 shows the proposed effect of fucoxanthin on extrinsic (death receptor) and intrinsic (mitochondrial) pathways of apoptosis. In the extrinsic pathway, the death ligands bind to transmembrane death receptors and trigger assembling on the cell surface. This aggregation employs adaptor proteins on the cytoplasmic site of the receptors, forming the death-inducing signaling complex (DISC) [141]. Following the formation of DISC, procaspase molecules are brought close to each other, enabling autocatalytic activation and release into the cytoplasm. A caspase cascade is activated, and caspase-8 mediates the cleavage of proapoptotic Bid into t-bid, which interacts with and causes the release of mitochondrial proapoptotic factors connecting the two pathways [142]. Fucoxanthin has been shown to induce procaspase-8 function [142]. In the intrinsic pathway, BAX and BID bind to the outer membrane of mitochondria. BAK then interacts with BAX and BID, causing the release of proapoptotic factors such as (Smac/DIABLO), HtrA2, OMI, AIF, and cytochrome c into the cytosol in response to diverse apoptotic stimuli. The binding of Omi/HtrA2 or Smac/DIABLO to an inhibitor of apoptosis proteins (IAPs) stimulates the activation of caspases and disrupts the interaction of IAPs with caspase-3 or -9. Fucoxanthin has been shown to inhibit XIAP and induce procaspases-3 and -9 functions [112]. Cytochrome c binds Apaf-1 triggering the formation of apoptosome and causing the activation of procaspase-9, leading to apoptosis [143].

#### 5.2.3. The Inhibition of PI3K/Akt Signaling Mediated Apoptosis Mechanisms by Fucoxanthin 

Several checkpoints in apoptosis expose a maintained equilibrium between cell death and cell survival in cells. Akt and NF-κB signaling pathways are associated with cell survival. The PI3K/Akt/protein kinase B (PKB) pathway is vital in selecting cellular processes, including proliferation, motility, cell growth, and survival in both tumor and normal cells [144]. This pathway imposes a significant assortment of intracellular events that can affect whether a cell will undergo apoptosis. Moreover, transcription factor NF-κB serves a critical role in many processes, including cell growth and survival, inflammation, and immune response [145]. Many agents, which induce apoptosis, such as TNF-α, induce activation of NF-κB [146]. The inhibition of NF-κB activation stimulates apoptosis [147]. Enhanced signaling of the PI3K/Akt pathway influences the transforming events in cancer. Akt, also known as PKB, controls several targets, including various apoptotic genes [132]. Akt is stimulated through recruitment to membranes by stimuli that provoke the production of phosphatidylinositol-3,4,5-triphosphates (PIP3) by phosphoinositide 3-kinase (PI3K).

Fucoxanthin has been shown to inhibit Akt directly in HG-induced renal fibrosis in GMCs [148]. Stimulated PI3K converts phosphatidylinositol-4,5-bisphosphate (PIP2) to PIP3 and activates downstream effectors such as mTOR and AKT. Akt can phosphorylate BAD on Ser136, dissociating BAD from the Bcl-2/Bcl-xL complex and losing its proapoptotic abilities. In HeLa cells, fucoxanthin increases Bad expression and decreases Bcl-2 expression [149]. HtrA2 is released to the cytosol from the mitochondria during apoptosis. However, as Akt phosphorylates HtrA2, apoptosis is inhibited.

In addition, Akt has been shown to phosphorylate Bax on serine residue 184 dependently [150]. The phosphorylation of Bax leads to a conformational change resulting in blocked activation. Moreover, Bim, an interactive mediator of cellular death, is a BH3 protein that Akt phosphorylates at serine residue, Ser87. The mitochondrial pathway of apoptosis is also activated by mTOR signaling that belongs to survival programs that are stimulated in cancers and promote cell survival by inhibiting apoptosis. Fucoxanthin has been shown to induce apoptosis in cervical cancer by inhibiting the PI3K/Akt/mTOR pathway [151]. mTOR acts as an inhibitor as rapamycin induces apoptosis [152]. Akt induces IKK, which then induces NF-κB, causing the activation of the Bcl-xL protein [153]. 

Furthermore, fucoxanthin was found to inhibit NF-κB by decreasing the stimulation of NF-κB and MAPKs in lipopolysaccharide-induced RAW 264.7 macrophages [154]. As the phosphorylation of proapoptotic proteins, Bim and Bax cause a decrease in proapoptotic potential, antiapoptotic factors Mcl-1 (induced myeloid leukemia cell differentiation protein) and XIAP (x-linked inhibitor of apoptosis protein) reduce both protein stability and anti-apoptotic properties. The phosphorylation of Mcl-1 and XIAP by Akt stimulates the degradation of the proteasomal structure, resulting in reducing Mcl-1 and XIAP protein expression [155]. In adult T-cell leukemia cells, fucoxanthin has been shown to reduce expressions of XIAP and Bcl-2 while activating procaspase-9, -8, and -3 [102].

The forkhead box 3 (FOXO 3) performs like a tumor suppressor in cancer. Its inactivation is correlated with the progression and initiation of cancer. FOXO transcription factors including FX1, FOXO3a, FOXO4, and FOXO6, are involved in apoptotic mechanisms. FOXO3a has been shown to upregulate Bim and Noxa. Increased expression of Noxa can enhance apoptosis of mitochondria by binding to Mcl-1, which can interrupt the antiapoptotic operations of protein Mcl-1. The phosphorylation of Akt in the PI3K signaling pathway is inhibited by FOXO3 [156]. The PI3K/AKT signaling pathway is a key controller of normal cellular processes involved in survival, proliferation, growth, apoptosis, and angiogenesis (Figure 5).

The proposed effect of fucoxanthin on PI3K/Akt signaling and mediated antiapoptotic modulations is depicted in Figure 5. The graph shows fucoxanthin inhibits AKT. PI3K converts phosphatidylinositol-4,5-bisphosphate (PIP2) to phosphatidylinositol-3,4,5-trisphosphate (PIP3) and stimulates downstream effectors such as mTOR and AKT. PI3K/Akt can phosphorylate and inhibit proapoptotic proteins (Bax and Bad). Fucoxanthin has been shown to induce the proapoptotic function of Bax [157]. Akt phosphorylates HtrA2, and apoptosis is inhibited. AKT activates mTOR, phosphorylating and stimulating the antiapoptotic protein MCL-1 (myeloid cell leukemia-1). AKT can activate NF-κB and IKK, causing the transcription of Bcl-xL (B-cell lymphoma-extra-large). Fucoxanthin has been shown to inhibit Bcl-xL [158]. AKT phosphorylates XIAP (X-linked inhibitor of apoptosis protein), then binds and inhibits caspases. Fucoxanthin has been found to inhibit XIAP function. Bim (Bcl-2-interacting mediator of cell death) and Noxa are inhibited by FOXO3 (Forkhead box 3), phosphorylated by AKT in the PI3K signaling pathway. Fucoxanthin has been shown to induce procaspase-3 function, leading to apoptosis. NF-κB, and members p52 and p100, play a role in stimulating the antiapoptotic protein Bcl-xL in the PI3k/Akt pathway. Studies show that inhibits NF-κB, p52, and p100, which may lead to the induction of apoptosis.

### 5.3. The Process of Angiogenesis

Like normal tissues, tumors need a sufficient number of metabolites, oxygen, and an efficient way to eliminate excretion [159]. In 1971, Judah Folkman proposed that tumors must have blood vessels to grow and survive, and with the discontinuation of this blood supply, cancer could be depleted into remission [160]. The requirements for survival vary amongst different types of tumors and can alter throughout tumor development. Tumor cells penetrate lymphatic or blood vessels, travel through the intravascular stream, and proliferate at another site by a metastatic process. The vascular network’s growth is crucial for cancer’s metastatic spread in tissue [161].

Angiogenesis is the development of new blood vessel growth that plays a vital role in forming a new vascular network. The vascular network supplies oxygen, nutrients, immune cells, and the removal of waste products, and in the absence of this network, tumors become apoptotic or necrotic [162]. The growth of tumors and metastases in numerous cancers is a significant factor in angiogenesis. Increased levels of angiogenesis correlate with decreased survival in cancer patients [163]. Cellular migration and invasion are essential to angiogenesis and cancer metastasis [164]. There are three mechanisms implied in endothelial cell migration: chemotaxis, haptotaxis, and mechanotaxis. Chemotaxis is the directional migration toward a chemokine source. Haptotaxis is the directed migration in response to the increasing concentration of ECM proteins. Mechanotaxis is the directional migration generated by mechanical forces. Normally, chemotaxis of cells involves growth factors, such as the basic fibroblast growth factor (bFGF) and vascular endothelial growth factor (VEGF). Haptotaxis is associated with increased endothelial cell migration, where integrins bind to the ECM component.

Studies have shown that angiogenesis in tumors is required for metastasis development and tumor development involving growth factors such as IL-8, VEGF (bFGF/FGF-2), and MMPs [165,166,167,168]. Overexpression of cytokines has damaging effects on angiogenesis and stimulates tissue invasion and migration. Pro-inflammatory cytokines, such as TNF-α, interleukin (IL)-1, IL-1, IL-6, and IL-8, along with cytokine growth factors including VEGF and FGF, establish a microenvironment that promotes the progression of tumors and stimulates processes such as angiogenesis, migration, and invasion [169].

#### The Effect of Fucoxanthin on Angiogenesis

The innovation of natural compounds as efficient angiogenesis inhibitors is a good approach to the prevention of cancer. The antiangiogenic effects of fucoxanthin were investigated in the rat aortic ring and cultured in human umbilical vein endothelial cells [170]. Fucoxanthin is known to strongly suppress the differentiation of endothelial progenitor cells into endothelial cells and the formation of new blood vessels [171]. In B16-F10 mouse melanoma cells, fucoxanthin (30 µM) suppressed invasion measured by a cell migration in wound healing as well as transwell invasion assay. Fucoxanthin also suppressed the union of B16-F10 cells to human umbilical vein endothelial cells that were activated by TNF-α. CD44, C-X-C, and matrix metallopeptidase 9 were also decreased, which are known to induce cancer invasion and migration and to reduce actin fiber formation in cells [172]. 

In vivo studies using mice engrafted with MDA-MB-231 cells and injected with 100 and 500 µmol/L of fucoxanthin showed antiangiogenic effects [121]. Fucoxanthin inhibited cell migration and tube formation significantly in a dose-dependent manner. The morphology of the HLEC cytoskeleton was transformed by fucoxanthin while decreasing pseudopod formation [119]. Angiogenesis and lymph angiogenesis are important phenomena involved in carcinogenesis and is linked to poor prognosis [173].

### 5.4. Cytokines in Cancer Development and Progression

In 1863, Rudolf Virchow first established the relationship between cancer and chronic inflammation by referring to immune cells introduced into tumor samples to interpret tumor development at sites of inflammation [174]. Inflammation mediates the promotion and initiation of metastasis, angiogenesis, and tumors [175]. In the tumor microenvironment, inflammation affects immune cells and activates cytokine-secreting. Cytokines are highly inducible secreted proteins that intervene with cell-to-cell communication of the immune system. They play an important role in controlling both cancer induction and protection. Cytokines are predominantly generated by macrophages and lymphocytes, though they can be manufactured by endothelial, connective tissue, epithelial cells, and adipocytes [176].

Consequently, it is not surprising that cytokines greatly influence tumor growth in vivo. This distinct and substantial group of anti- and pro-inflammatory factors are categorized into families based on their respective receptors or based on their structural homology. Alternatively, they represent systems with numerous functionally and molecularly unique members that are produced by cancer cells and function as inhibitors or promoters of tumor growth. Cytokines intervene as mediators of the effector response from inherited and attained cellular immunities, and they may engage in the mechanism of tumor cell circumvention of the immunosurveillance system [177]. Numerous cytokines modulate the inflammatory tumor microenvironment including TNF-α, TGF-β, IL-1, IL-6, IL-11, IL-19, IL-20, TGF-α, IL-23, and VEGF. These cytokines stimulate cytokine receptor activation leading to cancer cell invasion, proliferation, and intracellular signaling by NF-κB to progress tumor development [178].

#### The Effect of Fucoxanthin on Cytokines Formation and Release 

In bone marrow-derived immune cells, fucoxanthin reduced IL-1β, TNF-α, and IL-6 induced by LPS/ATP. Furthermore, fucoxanthin showed a decrease in IL-1β and IL-6 in LPS/ATO-induced astrocytes [179]. Fucoxanthin decreased TNF-α and IL-6 and increased IL-10 expression in BALB/c mice skin [180]. Moreover, fucoxanthin reduced IL-6, IL-1β, and TNF-α in LPS-induced sepsis mouse and cell inflammation models [181]. In acute lung injury, fucoxanthin reduced mRNA expression of proinflammatory factors such as IL-6, iNOS, Cox-2, and IL-10 and downregulated NF-κB signaling in monocyte/macrophage-like cells RAW264.7 [182]. IL-6 mRNA levels in white adipose tissue of diabetic/obese mice were decreased by fucoxanthin [183]. Meanwhile, fucoxanthin attenuated the production of IL-1β and IL-6 in human keratinocyte, HaCaT cells [184]. Fucoxanthin induced cytokine tumor necrosis factor-related apoptosis-inducing ligand (TRAIL) and stimulated apoptosis in cervical cancer cells via the PI3K/Akt/ NF-κB signaling pathway [185]. Studies show that fucoxanthin significantly reduced VEGF-C, VEGFR-3, NF-κB, phospho-Akt, and phospho-PI3K in HLEC in breast cancer [121]. Moreover, data showed that fucoxanthin inhibited mRNA expression of VEGF-C in MDA-MB-231 cells [121,186].

### 5.5. The Process of Autophagy

The naturally preserved process that is self-degradative and cleans any unnecessary or damaged components out of the cell is known as autophagy. Autophagy is also described as lysosomal degradation or “self-eating” [187]. It is upregulated in reactions to physiological conditions such as responses to diverse pathological stresses and starvation. A cell obliterates old or malfunctioning cellular components, which are reused to meet metabolic needs. Autophagy exerts cytoprotection by eliminating cytotoxic materials such as injured mitochondria and misfolded proteins [188]. Autophagy through the lysosomal degradation pathway represents types of stress including depletion of growth factors, hypoxia, and nutrient deprivation, which can lead to cell death [189]. Decreases in autophagic capacity produce malignant transformation and spontaneous tumors. In cancer, autophagy is considered to have both tumor-promoting and tumor-suppressive purposes and is known to be a double-edged sword. 

In tumors, autophagy has been found to limit promoters of cancer initiation, such as tissue destruction, inflammation, and genome instability. Therefore, the stimulation of autophagy could be valuable for cancer prevention. As tumor cells are exposed to stressors during metastasis, progression, and cancer therapy, autophagy is believed to enable tumor promotion through tumor cell survival. The survival response of cytoprotective autophagy offers cancer cells the ability to shield against starvation and escape apoptotic signals [190]. Autophagy-related genes (ATG) are a set of evolutionarily conserved genes that tightly regulate the production of autophagosomes [191]. Autophagosomes are double-membraned vesicles that contain cellular materials that are degraded through the process of autophagy. Autophagy is generally controlled by 11 ATG genes that have been identified in yeast which have orthologs in mammals. ATG proteins regulate autophagosome production through five stages: (1) initiation, (2) nucleation of the autophagosome, (3) elongation and expansion of the autophagosome membrane, (4) closure and fusion with the lysosome, and (5) degradation of intravascular cargo products. In tissue and mammary cells, autophagy plays a part in the differentiation and development of lumen structures called acini, as well as homeostasis. In the center of acini, mammary cells undergo apoptotic signaling following the shortage of extracellular matrix attachment. During this process, the reduction of ATG genes such as ATG5 and ATG7 in MCF10A cancer cells has been found to cause a decrease in clonogenic cells and increase pro-apoptotic caspase-3 [192].

#### 5.5.1. The Effect of Fucoxanthin on Autophagy

Many natural compounds have been evaluated in the modulation of autophagy. In MCF-7 breast cancer cells, cucurbitacin B induced autophagy and DNA damage by increasing the formation of ROS [193]. Fruit and vegetables with low asparagine have been found to block the fusion of autophagosomes with lysosomes and suppress lysosomal capabilities [194]. In breast cancer, berberine isolated from plants such as barberry, oregon grape, and tree turmeric induces autophagic cell death and apoptosis by activating beclin-1 and inhibiting the mTOR signaling pathway [195]. 

Fucoxanthin induces apoptosis and autophagy by upregulating beclin-1, caspase-3, and LC3 and downregulating Bcl-2 [196] (Figure 6). Fucoxanthin increased protein levels of LC3-II and beclin-1 while decreasing the phosphorylation of mTOR hepatocytes [197]. Fucoxanthin has been reported to stimulate apoptosis and/or autophagy in many cancer lines, such as nasopharyngeal cancer, HeLa, and SGC-7901 [198,199,200].

In Figure 6, PI3K is activated by extracellular growth factors. mTOR is an important target of autophagy and regulates the autophagy process. Fucoxanthin may inhibit mTOR directly. The Unc-51-like autophagy activating kinase (ULK1) and ATG13 participate in the initiation process and activate downstream factors. Beclin-1 and ATG14 form a complex and act in the autophagosome nucleation step. Fucoxanthin has been shown to induce Beclin-1, which induces the further formation of the autophagosome [200]. Next, ATG7 and ATG10 conjugate ATG5 to ATG12, and ATG7 and ATG3 join to protein 1 light chain 3 (LC3) [201]. The ATG5-ATG12 complex conjugates with ATG16 and forms the ATG5-ATG12-ATG16 complex, which binds and emerges on the autophagosomal membrane. ATG4 lipid oxidates LC3 to LC3-I, evoking the lipid phosphatidylethanolamine (PE). The elongation and expansion of the autophagosome membrane occur, and ATG7 and ATG3 catalyze LC3-I to form LC3-II. Fucoxanthin has been shown to induce LC3 and LC3-II directly [202]. Then, the autophagosome traffics to the lysosome and fuses to produce an autolysosome. Outer membrane-bound LC3-II is cleaved and recycled by ATG4, while lysosomal proteases degrade LC3-PE on cargo autophagosomes.

#### 5.5.2. Targeting Autophagy in Cancer: The Link between Autophagy and Apoptosis

There are two ways to directly link autophagy and apoptosis: (1) apoptosis could be controlled by autophagy, making the process more or less likely to occur, and (2) autophagy could be regulated by apoptosis by increasing or decreasing its performance. Overall, autophagy is a pro-survival mechanism allowing cell survival and prolonged starvation. The stimulation of autophagosome depletion has been introduced as a method of cell death due to the accumulation of autophagosomes and autolysosomes in the cytoplasm of dying cells, not including the activation of apoptosis [203]. In particular developmental environments, autophagy results in type II programmed cell death in cells expressing Bcl-xL or Bcl-2 or lacking Bax and Bak proteins [204].

There are many direct molecular links between autophagy and apoptosis which supports the idea of a link between the two processes. Atg5 stimulates DISC through interaction with FADD to initiate caspase-dependent cell death [205]. Beclin-1 interacts with the antiapoptotic protein, Bcl-2. As Bcl-2 and Beclin-1 are attached, Beclin-1 cannot activate autophagy. In apoptosis, FLICE-inhibitory protein (FLIP) blocks TRAIL-mediated cell death by meddling with caspase-8 activation. FLIP induces autophagy by inhibiting the Atg3-LC3 association. The conjugation of Atg12 to Atg3, vital for autophagosome formation, can regulate apoptosis. However, mutations of Atg3 prevent conjugation to Atg12, therefore inhibiting apoptosis. The p53 protein has the ability to promote or inhibit autophagy [206]. Studies show that in gastric SGC-7901 cells, fucoxanthin reduced cell viability, increased procaspase-3 and LC3, and downregulated Bcl-2 which effectively induces both apoptosis and autophagy [196].

### 5.6. Cancer and microRNA

MicroRNAs (miRNAs) are a group of small, single-stranded, endogenic, and non-protein-coding RNAs spanning 19 to 25 nucleotides [207]. According to the latest airbase database (https://www.mirbase.org/) (accessed on 8 June 2022), there are more than 35,000 hairpin precursor miRNAs identified in more than 270 organisms [208]. miRNAs silence genes by base-pairing with complementary sequences in 3’UTRs of their targets and regulate 30% of the protein-coding genome. They function in various biological processes, including tissue differentiation and organ development, cell proliferation, apoptosis, fat metabolism, and insulin secretion [209]. Under normal physiological conditions, miRNAs function in feedback mechanisms by protecting major biological processes such as apoptosis, cell proliferation, and differentiation [210]. The deregulation of a single or small subset of miRNAs was stated to have a profound effect on the expression of hundreds of mRNAs. Deregulated miRNA expression is associated with tumor progression, development, and response to therapy [211]. In cancer, miRNAs play a role in metastasis, oncogenesis, and resistance to therapy and can be classified as oncogenes (oncomirs) or tumor suppressor genes [212]. Oncogenic miRNAs or oncomirs play a crucial role in the initiation and progression of human cancer. They are overexpressed in many cancer types by acting through various downstream targets [213]. Tumor suppressor miRNAs play a crucial role in the downregulation of cancer cells through inhibiting epithelial–mesenchymal transition (EMT), cell proliferation, oncogene expression, and the promotion of apoptosis [214].

#### The Effect of Natural Compounds and Fucoxanthin on miRNA

Resveratrol is isolated from various sources, such as mulberry grapes and peanuts. In MCF-7 breast cancer cells, resveratrol modulates the p53 signaling pathway and has antitumor effects by enhancing tumor suppressors miR-424, miR-503, and miR-34a [215]. Genistein downregulates miR-155 breast cancer cells and leads to the Induction of cell death along with an increase in p21, PTEN, FOXO3, and suppression of cell viability [216]. Docosahexaenoic acid (DHA) is ample in seaweed oil and is a key component of infant formula for vision and intelligence. DHA treatment in MCF-7 cells increases miR-23b, miR-320b, and miR27b secretion and can modulate miRNAs and attenuate tumor angiogenesis [217].

In leukemia cells, 6-gingerol upregulates miR-27b, reduces the PPARγ-NF-κB pathway, and induces apoptosis in tumor cells [218]. Berberine suppresses endometrial cancer by inducing tumor suppressor miR-101. In ovarian cancer, combination therapy of berberine and cisplatin downregulates miR-93 and reduces cell death leading to G0/G1 cell cycle arrest [219]. Aplysin, a marine bromide natural compound, induces downregulation of mitogen-activated protein kinase kinase 1 (MEK1) through upregulation of miR-181 [220]. In prostate cancer cells, ellagitannin and its derivatives isolated from *Balanphora japonica makino* downregulate miR-21 expression resulting in the upregulation of tumor suppressor FOXO3a and PTEN while downregulating Wnt/β-catenin and Akt phosphorylation [221]. In MDA-MB-231 TNBC cells, glyceollins isolated from soybean have been found to downregulate miR-185 and reduce metastasis formation [222]. A combination of indole-3 carbinol and diindolylmethane isolated from vegetables such as cauliflower, kale, cabbage, and broccoli upregulate in prostate cancer cells miR-150-5p, modulating MAPK pathways and inhibiting cell viability and invasion [223]. In MDA-MB-231 and MCF-7 breast cancer cells, fucoidan isolated from brown seaweed reduces miR-17-5p expression and regulates the PTEN pathway. Fucoidan has also been shown to epigenetically control the invasion and migration of miRNA-29c, leading to the control of metastasis and tumorigenesis [224].

Few reports on the effect of fucoxanthin in miRNA were described in the literature. In LPS-stimulated RAW 256.7 macrophages, fucoxanthin decreased levels of expression of mir-146b compared to LPS treatment [225]. The upregulation of mir-146b is related to inflammatory diseases, including inflammatory bowel disease [226] (Figure 7). 

Due to miRNA’s ability to regulate the gene expression pattern, miRNA also simultaneously controls many cellular pathways [227]. The miRNAs are promising molecules that serve as biomarkers for diagnostic, prognostics, and cancer therapeutics. Natural products have fewer adverse effects on normal cells and unveil antitumor character by targeting many cellular signaling pathways [228]. Likewise, miRNAs regulate many biological processes such as cell death, tumor growth, and progression [229]. Both natural products and miRNAs display an impact on multiple cellular targets. There is great interest in the notion that natural products could modulate miRNAs and pave the way for cancer treatment.

Figure 7 shows the proposed effect of fucoxanthin on miRNA. The figure shows that miRNAs are transcribed by RNA polymerase II, producing pri-miRNAs. Pri-miRNAs are processed into pre-miRNA by gene 8 in the critical region of Drosha and DiGeorge syndrome (DGCR8). Pre-miRNAs are then transported to the cytoplasm by exportin 5 and Ran-GTP. RNAse-III types of enzyme, TAR RNA binding protein (TRBP), and Dicer act in cleaving pre-miRNA into the miRNA duplex. miRNAs are assembled into the miRNA-induced silencing complex (RISC) with the assistance of argonaut protein (AGO), producing mature miRNA. The miRNA is then degraded, cleaved, or mediated by translational stimulation. Fucoxanthin has been shown to decrease levels of mir-146b expression [230].

## 6. Summary

The literature reviewed indicates that fucoxanthin has many anticancer effects, including antiproliferative, antiangiogenic, anti-migration, anti-invasion, and anti-metastatic. Moreover, the literature shows fucoxanthin’s ability to inhibit cytokines and growth factors such as TNF-α and VEGF, which stimulate the activation of downstream signaling pathways such as PI3K/Akt autophagy and pathways of apoptosis in many cancer cell lines (Figure 8). These pathways are essential in proliferation, cell cycle arrest, angiogenesis, migration, invasion, metastasis, and apoptosis. Fucoxanthin can modulate these pathways and thus provides a significant role in cancer regression. The activation of the PI3K/Akt pathway through stimulation of proangiogenic cytokine VEGF and binding to VEGFR-3 can activate autophagy. Fucoxanthin may inhibit mTOR and further cause stimulation of autophagy proteins such as Atg5 and Atg12, which interacts with apoptosis domain FADD stimulating caspase cell death. Extrinsic and intrinsic pathways of apoptosis are activated by stimulating TNF-α and by binding to TNFR1 or FASL.

Fucoxanthin may cause inhibition of Bcl-xL, thereby leading to activation of apoptosis through stimulation of caspases-8, -3, and -9. The activation of the PI3K/Akt pathway through stimulation of proangiogenic cytokine VEGF and binding to VEGFR-3 can also activate apoptosis. Fucoxanthin may reduce XIAP function, a regulator of extrinsic or intrinsic apoptosis. Fucoxanthin may cause apoptotic effects by inhibiting NF-κB members p52 and p100, which play a role in stimulating the antiapoptotic protein Bcl-xL in the PI3k/Akt pathway. Invasive cancer metastasis is related to the proliferation and deregulation of the cell cycle. Fucoxanthin may inhibit proliferation and cause induction of G_0_/G1, G1, and G2 cell cycle arrest. Oncogenic miRNA initiates various downstream targets and is overexpressed in cancer, while tumor suppressor miRNAs suppress cancer cells and inhibit cancer progression. Fucoxanthin may inhibit oncogenic miR-155 and stimulate tumor suppressor miR-146b, leading to the inhibition of cancer progression. 

The literature reviewed indicate that fucoxanthin anticancer effects may be related to TNF-α, tumor necrosis factor-α, TNFR1, tumor necrosis factor receptor 1; VEGF, vascular endothelial growth factor; VEGF-C, vascular endothelial growth factor-C; VEGFR-3, vascular endothelial growth factor receptor-3; FasL, Fas ligand; ATG, autophagy-related genes; ULK1, Unc-51-like autophagy activating kinase 1; FADD, Fas-associated death domain; TRADD, TNF receptor-associated death domain; DISC, death-inducing-signaling complex; Bid, BH3-interacting domain death agonist; tBid, truncated Bid; Bcl-2, B-cell lymphoma-2; Bax, Bcl-2-associated X protein; Bad, Bcl-2-associated death promoter; Bcl-xL, B-cell lymphoma-extra-large; Noxa, BH3-protein family member; XIAP, X-linked inhibitor of apoptosis protein; PI3K, phosphoinositide 3-kinase; PIP2, phosphatidylinositol-4,5-bisphosphate; PIP3, phosphatidylinositol-3,4,5-trisphosphate; Akt, protein kinase B; NF-ĸB, nuclear factor-ĸB; mTOR, mammalian target of rapamycin; FOXO, forehead box; and IKK, an inhibitor of NF-κB kinase.

## 7. Conclusions

Cancer is associated with aberrant signaling that can lead to proliferation, angiogenesis, metastatic invasion and migration, invasive metastasis, and tumor growth progression. Many experimental studies have revealed that fucoxanthin has many anticancer effects, including antiproliferative, antiangiogenic, anti-migration, anti-invasion, and anti-metastatic. This compound modulates miRNA and induces cell cycle growth arrest, apoptosis, and autophagy in many cancer cell lines. Upon ingestion, fucoxanthin is broken down into fucoxanthinol in the gastrointestinal tract and is converted into amarouciaxanthin A in the liver. The use of fucoxanthin is low due to its poor stability, low water solubility, and limited bio-accessibility. Experimental research has demonstrated that the utilization of fucoxanthin was assumed to speed up metabolism. This review highlights the different critical mechanisms by which fucoxanthin inhibits diverse types of cancer, including breast, prostate, gastric, lung, and bladder development and progression. Fucoxanthin has been shown to inhibit cell proliferation due to cell growth arrest at G_1_ and G_0_/G_1_ phases and exerts antiproliferative effects on several cell lines of breast cancer cells. In breast cancer cells MCF-7, fucoxanthin induced apoptosis and offered protection against DNA damage. Moreover, fucoxanthin shows antiproliferative effects on urinary bladder cancer, melanoma, prostate cancer, gastric cancer (MGC-803), and breast cancer cells. In mouse melanoma cells, fucoxanthin suppressed invasion quantified by cell migration in wound healing and transwell invasion assay. Moreover, fucoxanthin showed antiangiogenic effects by suppressing adhesion in B16-F10 melanoma cells. In gastric cancer cells MGC-803, fucoxanthin induced cell cycle arrest and apoptosis at the G2/M phase. Fucoxanthin also inhibited AP-1, Akt, and NF-κB activation and decreased antiapoptotic proteins XIAP and Bcl-XL in lymphoma cancer cells. This review re-examines the current literature and provides critical supportive evidence for fucoxanthin’s possible therapeutic use in cancer. Therefore, the available evidence in this review indicates that fucoxanthin might be a potential candidate in cancer therapy with multiple anticancer targets. 

## Figures and Tables

**Figure 1 ijms-23-16091-f001:**
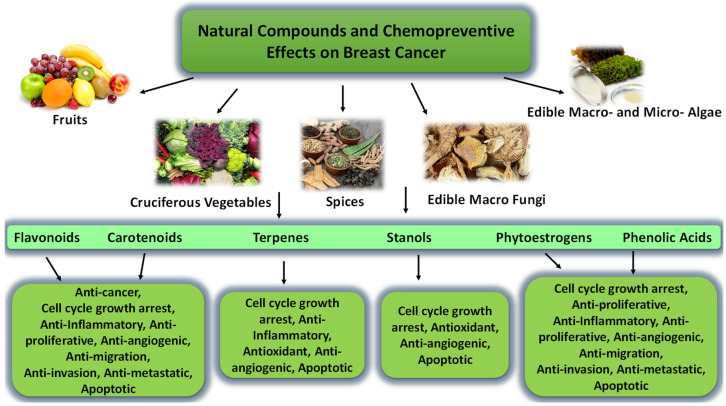
Natural compounds and their chemopreventive effects on cancer.

**Figure 2 ijms-23-16091-f002:**
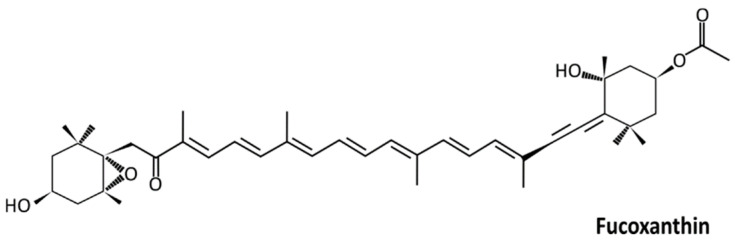
The chemical structure of fucoxanthin [89].

**Figure 3 ijms-23-16091-f003:**
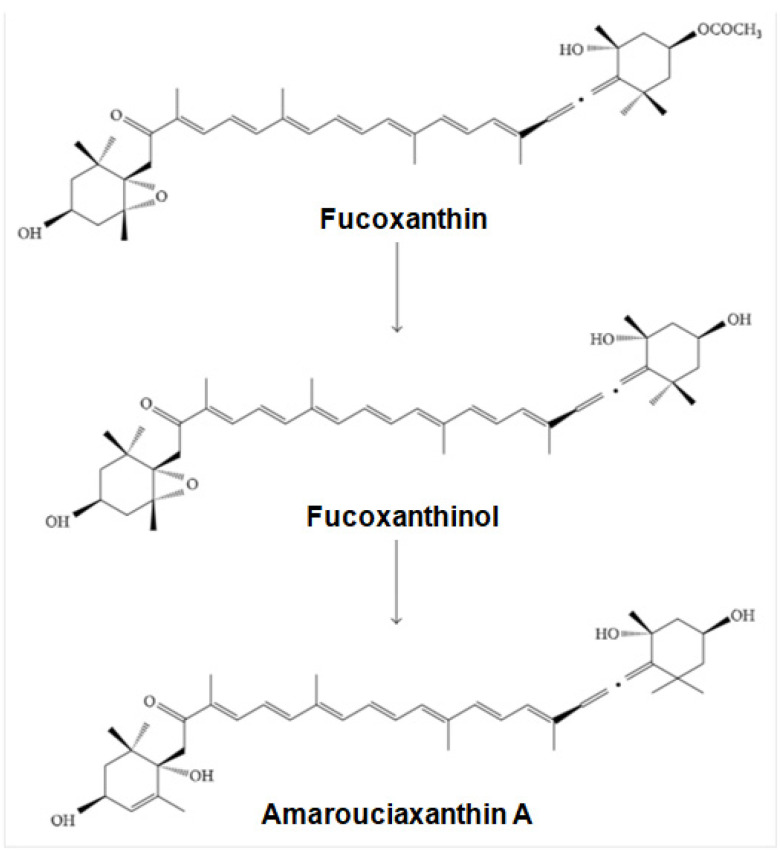
Fucoxanthin metabolism to fucoxanthinol and amarouciaxanthin A [100].

**Figure 4 ijms-23-16091-f004:**
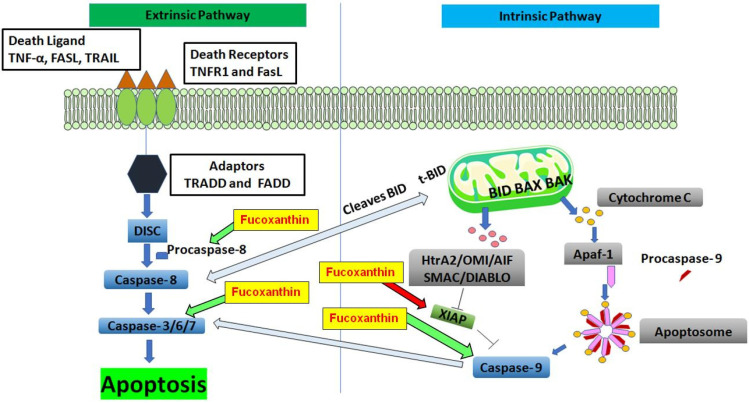
The proposed effect of fucoxanthin on the extrinsic (death receptor) and intrinsic (mitochondrial) pathways of apoptosis. Red arrows indicate that inhibition is induced by fucoxanthin, and green arrows indicate activation is induced by fucoxanthin.

**Figure 5 ijms-23-16091-f005:**
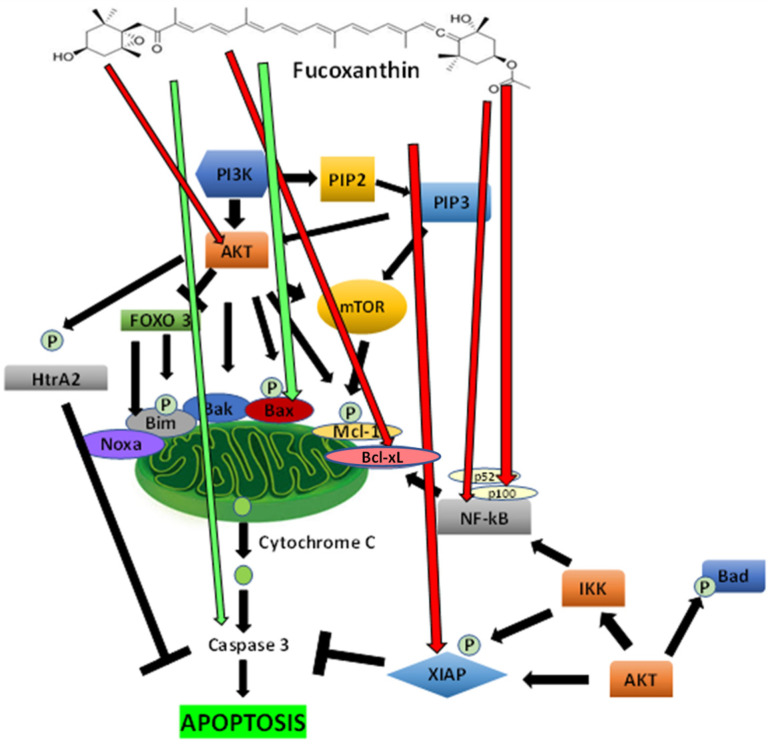
Proposed effects of fucoxanthin on PI3K/Akt signaling and mediated antiapoptotic regulation. Red arrows indicate that inhibition is induced by fucoxanthin, and green arrows indicate activation is induced by fucoxanthin.

**Figure 6 ijms-23-16091-f006:**
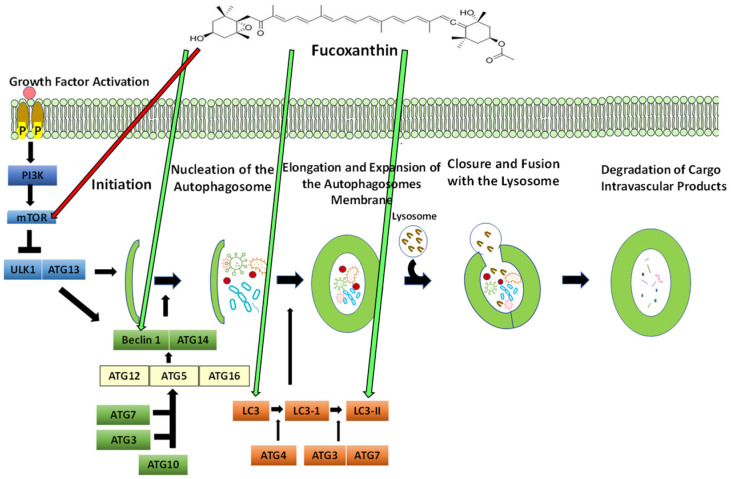
The proposed effect of fucoxanthin on autophagosome production. Red arrows indicate that inhibition is induced by fucoxanthin, and green arrows indicate activation is induced by fucoxanthin.

**Figure 7 ijms-23-16091-f007:**
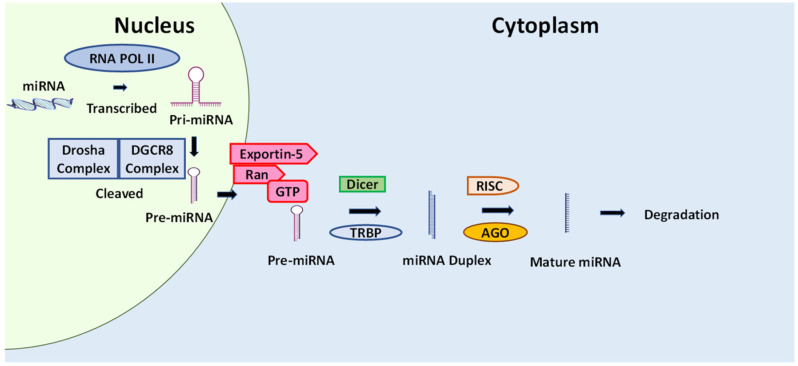
The proposed effect of fucoxanthin on miRNA. The red arrow indicates that inhibition is induced by fucoxanthin. miRNAs are developed from pri-miRNAs that are transcribed from other independent miRNA genes.

**Figure 8 ijms-23-16091-f008:**
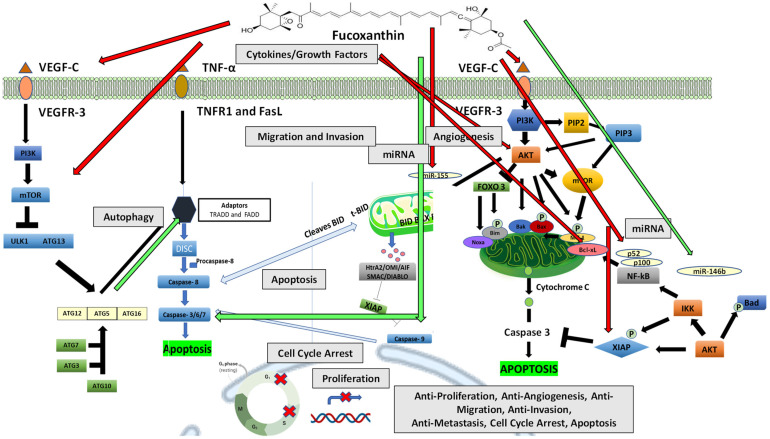
Proposed fucoxanthin action on cancer progression and development by modulating diverse signaling transduction pathways. Red arrows indicate that inhibition is induced by fucoxanthin, and green arrows indicate activation is induced by fucoxanthin.

**Table 1 ijms-23-16091-t001:** Relative survival rates and percentages by cancer type [15,16,17,18,19,20,21,22,23,24,25,26,27,28].

Cancer Type	Survival Rate	% of Survival Rates
**Breast cancer**	5-year survival rate	96%
**Non-invasive breast cancer**	10-year survival rate	84%
**Lung cancer**	5-year survival rate	17.7%
**Bladder cancer**	5-year survival rate	77%
**Prostate cancer**	5-year survival rate	98%
**Cutaneous melanoma**	5-year survival rate	26.4%
**Leukemia**	10-year survival rate	87%
**Colorectal cancer**	5-year survival rate	64.4%

## Data Availability

Not applicable.

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
