# Peer review of "Anticancer Effects of Fucoxanthin through Cell Cycle Arrest, Apoptosis Induction, Angiogenesis Inhibition, and Autophagy Modulation"

_ijms, 2022, doi:10.3390/ijms232416091_

Round 1

Reviewer 1 Report (Previous Reviewer 3)

This manuscript is not deserved for its publication in the present form. Some points should be improved and explained by the authors.

1.       Authors should maintain uniformity in the spelling of beta or alpha prefixes. Either they should leave them as words or write them with the Greek letter symbol.

2.       Chemically, it is better to define carotenes as hydrocarbons that are isoprene derivatives and xanthophylls as oxygen derivatives of carotenes instead of the sentence – “Carotenes have no oxygen and are classified as hydrocarbons, and xanthophylls have oxygen.” (Lines 206-207)

3.       What did the authors mean when they wrote down the following sentence - "Its acidic pKa value is 14.03, and its pKa value is -2.7." ? (Lines 229-230)

Author Response

XXXXXXXXXXXXXXXXXXXXXXXXXXXXXXXXXXXXXXXXXXXXXXCCCCCCCCCCCCCCCC

Dear Academic Editor:

We appreciate the reviewer's suggestions to improve our manuscript. We revised the previous version of the Manuscript ID: IJMS-1968918, "Anticancer Effects of Fucoxanthin through Cell Cycle Arrest, Apoptosis Induction, Angiogenesis Inhibition, and Autophagy Modulation."

Reviewer 1

  1. Authors should maintain uniformity in spelling beta or alpha prefixes. Either they should leave them as words or write them with the Greek letter symbol.

Response: All spellings of beta and alpha prefixes have been uniform throughout the review and changed to the Greek letter symbols α and β under section 4. Natural Products, Section 4.1 Carotenoids and Fucoxanthin (Line 204), and 5. Anticancer Mechanisms of Fucoxanthin, Section 5.2.1 The Intrinsic and Extrinsic Pathways of Apoptosis (Line 339).

  1. Chemically, it is better to define carotenes as hydrocarbons that are isoprene derivatives, and xanthophylls as oxygen derivatives of carotenes instead of the sentence – "Carotenes have no oxygen and are classified as hydrocarbons, and xanthophylls have oxygen." (Lines 206-207).

Response:  Thank you for your recommendation. Carotenes and Xanthophylls have been correctly defined and updated under section 4. Natural Products, Section 4.1 Carotenoids, and Fucoxanthin, (Lines 206-207).

  1. What did the authors mean when they wrote down the following sentence - "Its acidic pKa value is 14.03, and its pKa value is -2.7."? (Lines 229-230)

Response: This sentence has been corrected and changed to Fucoxanthin acidic pKa value is 14.03, and its basic pKa value is -2.7. 4 under section 4. Natural Products, Section 4.1 Carotenoids and Fucoxanthin (Lines 229-230).

Reviewer 2 Report (Previous Reviewer 2)

The work can be published in IJMS.

Author Response

No response needed

Reviewer 3 Report (New Reviewer)

In this article the authors review the effects of fucoxanthin in cell cycle arrest, apoptosis, angiogenesis and autophagy considering its particular anticancer activities. This review has its relevancy in the field of novel approaches in cancer therapy as it well described the biochemistry and the role of fucoxanthin in the biological processes which result modulated in cancer progression. The article is generally well written and exhaustive in the points treated.
Here are summarized some point to be considered for the revision:

·     2nd paragraph: “Cancer statistics”. Adding a table resuming the percentage of statistic discussed in the paragraph would make more effective the reading.

·    Figure 1: For a better understanding of the role of each natural compounds I would suggest reorganizing the figure specifying for every component their natural compounds and the relative effect as it is written in the text
·     Line 271: please write ‘in vivo’ in cursive
·    Line 335: please write ‘Figure4’ in bold, as the figure were cited this way in all the text
·    Lines 333-334 and lines 389-390 repeat the same sentence
·   Lines 399-400 Fucoxanthin has been shown to induce procaspase 8 function, please add citation

·Lines 481-482 Fucoxanthin has been shown to induce the proapoptotic function of Bax, please add citation
·        Lines 485-496 Fucoxanthin has been shown to inhibit Bcl-xL, please provide a citation
·    Lines 489 to 493 indicate two opposite effects of fucoxanthin on the same targets, please clarify
·    Lines 518-519 reports about studies but cite just one, please provide more than one citation
·      Line 582-583 please provide citation
·      Figure 6: the blue boxes are empty, please revise the figure.
·     Lines 645-646 please add a citation
·      Lines 651-652 please add a citation
·    Paragraph Targeting Autophagy in Cancer: The Link Between Autophagy and Apoptosis. The authors do not mention the role of Fucoxanthin in this link. Can the authors add some information about the role of the compound in the link of the two processes?

· Reference 221 is about the role of miR185: in the text there are not any mentions of this miRNA, please consider mentioning miR185 in the text.

Author Response

Dear Academic Editor:

We appreciate the editor and reviewer's suggestions to improve our manuscript. We have addressed all of the reviewer's comments in the Manuscript ID: ijms-1968918, "Anticancer Effects of Fucoxanthin through Cell Cycle Arrest, Apoptosis Induction, Angiogenesis Inhibition, and Autophagy Modulation" as follows:

  1. 2ndparagraph: "Cancer statistics." Adding a table resuming the percentage of statistics discussed in the paragraph would make the reading more effective.

Response: Thank you for your suggestion. A table was added for the percentage of statistics under Section 2: Cancer Statistics.

  1. Figure 1: For a better understanding of the role of each natural compound, I would suggest reorganizing the figure specifying for every component their natural compounds and the relative effect as it is written in the text

Response: Figure 1 under section 4: "Natural Compounds" has been updated, specifying each natural compound and its associated effects, as requested.

  1. Line 271: please write 'in vivo' in cursive

Response: As requested, 'in vivo' has been written in cursive on lines 273-274.

  1. Line 335: please write 'Figure 4' in bold, as the figure was cited this way in all the text

Response: Thank you for your observation. Line 337, Figure 4 has been written in bold.

  1. Lines 333-334 and lines 389-390 repeat the same sentence

Response: The repetitive sentence has been removed from lines 391-392.

  1. Lines 399-400 Fucoxanthin has been shown to induce procaspase 8 function; please add a citation

Response: A citation has been added to line 401.

  1. Lines 481-482 Fucoxanthin has been shown to induce the proapoptotic function of Bax; please add a citation.

Response: A citation has been added to line 482.

  1. Lines 485-486 Fucoxanthin has been shown to inhibit Bcl-xL; please provide a citation

Response: A citation has been added to line 486.

  1. Lines 489 to 493 indicate two opposite effects of fucoxanthin on the same targets; please clarify

Response: Lines 489-493 have clarified and explained proteins' apoptotic and antiapoptotic functions.

  1. Lines 518-519 reports aboutstudies but cite just one; please provide more than one citation

Response:  More citations (3) have been added to lines 518-520.

  1. Line 582-583, please provide a citation

Response: A citation has been added to line 583.

  1. Figure 6: the blue boxes are empty; please revise the figure.

Response: The figure was updated. The blue boxes show PI3K, mTOR, ULK1, and ATG13 in Figure 6.

  1. Lines 645-646, please add a citation

Response: A citation has been added to line 645.

  1. Lines 651-652, please add a citation

Response: A citation has been added to line 651.

  1. Paragraph Targeting Autophagy in Cancer: The Link Between Autophagy and Apoptosis. The authors do not mention the role of fucoxanthin in this link. Can the authors add some information about the role of the compound in the link of the two processes?

Response: The role of fucoxanthin involvement in the link between autophagy and apoptosis has been added to paragraph 2 of section 5.5.2, Targeting Autophagy in Cancer: The Link Between Autophagy and Apoptosis. Lines 677-680.

  1. Reference 221 is about the role of miR185: in the text, there are not any mentions of this miRNA; please consider mentioning miR185 in the text.

Response: Information on miR-185 has been added to paragraph 2 of section 5.6.1, The Effect of Natural Compounds and Fucoxanthin on miRNA, Lines 725-726.

This manuscript is a resubmission of an earlier submission. The following is a list of the peer review reports and author responses from that submission.

Round 1

Reviewer 1 Report

This manuscript addresses the use of fucoxanthin as an anti-breast cancer agent. However, I had very much difficulty following the realistic impact and mechanisms based on which these claims would be substantiated.

Major points:

- the introduction into the review is very flatly written and does not take into account the heterogenous nature and different subtypes of breast cancer

- moreover, the treatment section of this is written up without any awareness of real life therapy approaches to breast cancer and makes to general claims in order for this manuscript to be valid

- the aim of this review is in no way clearly defined and as such the review lacks clarity

- the review then proceeded further and evaluates some pathways and the impact of fucoxanthin, prior to establishing the impact of these pathways on breast cancer, thus leaving the topics highly disconnected

I would suggest to either decide on a specific subtype of BC and go more in-depth on these topics and pathways. Also consider, what clinically applicable markers could be tested for application of this treatment as we know that not all cancers will respond in the same way

Minor points:

- the abstract needs to be rewritten and shortned

All in all I would not recommend this manuscript for publication in its current form - there needs to be major rewriting efforts to coherently connect breast cancer carcinogenesis to the proposed treatment.

Author Response

cccccccccccc

Response to Reviewer 1:

  1. The introduction to the review is very flatly written and does not take into account the heterogeneous nature and different subtypes of breast cancer

Response: We agree with the reviewer's comments, and a new section named "Subtypes of breast cancer" (Section 2) was added to the review. In this section, we are discussing the heterogenous nature and different subtypes of breast cancer, including luminal ER-positive (luminal A and Luminal B), human epidermal receptor factor -2 (HER2) enriched, and basal-like, as well as triple-negative breast cancer.

  1. Moreover, the treatment section of this is written up without any awareness of real-life therapy approaches to breast cancer and makes general claims in order for this manuscript to be valid

Response: We agree with the reviewer, and as suggested, we have updated Section 3 - Treatment options for breast cancer, where we are discussing all the current treatment options for breast cancer.

  1. The aim of this review is in no way clearly defined, and as such, the review lacks clarity

Response: Our review focuses on the effect of fucoxanthin on different pathways associated with cancer. As suggested by the reviewer, we included the description of several pathways that lead to tumor progression before discussing the molecular mechanisms of fucoxanthin in breast cancer inhibition. Due to the limited information and published manuscripts on the effects of fucoxanthin on breast cancer, we also discussed the effect of fucoxanthin on other cancer types. The updated review includes the addition of cellular pathways on breast cancer to further understand breast cancer pathophysiology, the addition of clinically applicable markers, and modifications in sections to better understand the cellular processes of breast cancer. As stated below, in response to suggestion #4, we have added the listed additional sections. Also, the aim of the review was updated in the Abstract and Introduction (paragraph 3 - lines 12- 15).

  1. The review then proceeded further and evaluated some pathways and the impact of fucoxanthin prior to establishing the impact of these pathways on breast cancer, thus leaving the topics highly disconnected.

Response: We agree with the reviewer, and as suggested, we have added more information to show the impact of these pathways on the progression of breast cancer. We are discussing the relevance of each pathway prior to the treatment with fucoxanthin, which helps to demonstrate the efficacy of this compound in inhibiting breast tumor development. The following sections have been added: - 5.1 Cell Proliferation and Cell Cycle Arrest Mechanisms, - 5.2.1 The Intrinsic and Extrinsic Pathways of Apoptosis, - 5.3. The process of angiogenesis, - 5.4 Cytokines in Breast Cancer Development and Progression, - 5.5 The process of autophagy.

  1. I would suggest either deciding on a specific subtype of BC and going more in-depth on these topics and pathways. Also, consider what clinically applicable markers could be tested for application of this treatment as we know that not all cancers will respond in the same way

Response: As suggested by the reviewer, the manuscript was revised in order to go more in-depth and help to clarify the role of several pathways in breast cancer progression. We chose to highlight the effect of fucoxanthin in multiple breast cancer subtypes due to limited published articles and research on fucoxanthin and breast cancer. We have also added clinically applicable markers under section 2 (Subtypes of breast cancer, paragraph 1 - lines 13, 20, and 21) and Section 3 (Treatment options for breast cancer, paragraph 1, lines 15-17). We have also described fucoxanthin's anti-cancer effects on NF-κB members p52 and p100, which are used as clinical markers in breast cancer, especially in estrogen-independent breast cancers, under the section, 5.2.2 (The Effect of Fucoxanthin on Apoptosis paragraph 2, (line 8).

  1. The abstract needs to be rewritten and shortened

Response: We agree with the reviewer, and the abstract has been revised and shortened.

  1. All in all, I would not recommend this manuscript for publication in its current form - there need to be major rewriting efforts to connect breast cancer carcinogenesis to the proposed treatment coherently.

Response: According to the reviewer's suggestions, we have updated this review to focus on the effects of fucoxanthin by discussing cellular processes such as cell proliferation, cell cycle arrest, apoptosis, PI3K-AKT, angiogenesis, and autophagy, which are involved in breast cancer progression. By discussing the mechanisms that contribute to breast tumor progression, we are providing a better understanding of the molecular mechanisms of fucoxanthin, which could provide a better option for breast cancer treatment.

Author Response

Response to Reviewer 2:

  1. Figure 3 is incorrect. Scheme 1 shows the Figure from ref., and it is different from this in paper, but it is not completely right.

Response: We appreciate and agree with the reviewer. Figure 3 under Section 4.4 "Bioavailability and Metabolism of Fucoxanthin," has been updated.

  1. In line 156-157 pro-vitamin A or provitamin A; non-vitamin A or non-provitamin A

Response: The content of these lines has been updated. We have made changes to "provitamin A and non-provitamin A," under section 4.2 Carotenoids (lines 9, 10, and 11).

  1. Many other things should be unified like Bcl-xl or Bcl-XL, bold and cap letters in figures….

Response: We appreciate and accept the reviewer's suggestions. Bcl-xL has been unified through the text, and the figures were updated regarding the bold and cap letters.

Reviewer 3 Report

The manuscript is interesting. The Authors have presented the introduction well and planned the various sections of the manuscript properly. In addition to pointing out standard treatments for breast cancer, they highlighted bioactive compounds found in natural products that can potentially prevent the growth of cancer cells focusing on carotenoids and especially fucoxanthin. Below are some comments on the reviewed manuscript.

1. Lines 159-160: The phrase beta-carotene appeared twice in the presented sentence.

2. Lines 168-169: The name of microalgae that produce fucoxanthin should be in italics.

3. Lines 191-192: Please provide information about the source according to which the Authors posted the information about the acidity constant of fucoxanthin ("Its acidic pKa 191 value is 14.03, and its basic pKa value is -2.7").

4. The formulas of fucoxanthin and its metabolites shown in Figure 3 need improvement. In the formula of fucoxanthin, some of the bonds in the cyclic ring at the carbon atoms contain substituents located in the same conformation. With a tetrahedral carbon atom one substituent is located in front of the plane the other behind the plane. Besides, in the formulas of fucoxanthinol and amarouciaxanthin A, the OH group is missing at one of the rings.

5. Lines 242-245: The Authors, when listing the effects of fucoxanthin, provided only one literature reference. Please provide 2 more.

6. The names of journals in the Reference section should be capitalized.

Author Response

Response to Reviewer 4:

  1. Lines 159-160: The phrase beta-carotene appeared twice in the presented sentence.

Response: The sentence was updated under section 4.2 Carotenoids (paragraph 1, line 10).

  1. Lines 168-169: The name of microalgae that produce fucoxanthin should be in italics.

Response: The name of the microalgae that produce fucoxanthin has been italicized under section 4.3 Fucoxanthin (paragraph 1, lines 2 and 3).

  1. Lines 191-192: Please provide information about the source according to which the Authors posted the information about the acidity constant of fucoxanthin ("Its acidic pKa 191 value is 14.03, and its basic pKa value is -2.7").

Response: According to the reviewer's request, a reference source has been added to section 4.3 Fucoxanthin (paragraph 2, line 3, where we have the information about the acidity constant of fucoxanthin.

  1. The formulas of fucoxanthin and its metabolites are shown in Figure 3 need improvement. In the formula of fucoxanthin, some of the bonds in the cyclic ring at the carbon atoms contain substituents located in the same conformation. With a tetrahedral carbon atom, one substituent is located in front of the plane, the other behind the plane. Besides, in the formulas of fucoxanthinol and amarouciaxanthin A, the OH group is missing at one of the rings.

Response: We appreciate the reviewer's comments. The formulas of fucoxanthin and its metabolites are shown in Figure 3, under Section 4.4 Bioavailability and Metabolism of Fucoxanthin, which has been updated.

  1. Lines 242-245: The Authors, when listing the effects of fucoxanthin, provided only one literature reference. Please provide 2 more.

Response: 5. As recommended by the reviewer, two more references have been added under Section 5.2, The Effect of Fucoxanthin on Cell Proliferation and Cell Cycle Arrest (paragraph 1, line 5).

  1. The names of journals in the Reference section should be capitalized.

Response: As requested, the names of journals in the reference section have been capitalized.

Round 2

Reviewer 2 Report

The review can be published in IJMS.

Author Response

no response is needed

Reviewer 3 Report

The manuscript in its present form is suitable for publication.

Author Response

no response is needed.

Round 3

Reviewer 2 Report

Paper can be published.

Author Response

no response is needed.